# Coupling Evaluation and Spatial–Temporal Evolution of Land Ecosystem Services and Economic–Social Development in a City Group: The Case Study of the Chengdu–Chongqing City Group

**DOI:** 10.3390/ijerph20065095

**Published:** 2023-03-14

**Authors:** Qikang Zhong, Zhe Li, Yujing He

**Affiliations:** School of Architecture and Art, Central South University, Changsha 410083, China

**Keywords:** Chengdu–Chongqing city group, land ecosystem services, economic–social development, coupling and coordination, spatial and temporal evolution

## Abstract

The Chengdu–Chongqing city group (CCCG) is one of the regions with acute human–land conflicts in China at present. The current rapid development of CCCG has caused a large negative impact on regional land ecosystem services. Land ecosystem is the basis of economic development. Reasonable economic development is also the inherent requirement of land ecosystem and an important support for land ecosystem protection. Promoting the coordinated development of the economy and land ecosystems is a prerequisite for this city group to achieve ecological protection and high-quality development. Taking CCCG as an example, this paper constructs a coupling evaluation model of economic–social development and land ecosystem service by using the entropy weight method, coupling coordination degree model, gravity center model, and standard deviation ellipse model, and explores the coupling coordination degree and spatial–temporal evolution rule of the above two systems. The study found that, from 2005 to 2020, the overall economic–social development of the CCCG showed an upward trend with a regular pattern of a ‘High in the east and west, low in the central’, and the ‘dual-core’ spatial structure with Chengdu and Chongqing as the core ; the overall performance of land ecosystem services had a gentle slope downward trend with a ‘U’-shaped spatial pattern of “high around and low in the middle”. The results also show that the coupling coordination degree of economic–social development and land ecosystem services in CCCG continues to rise steadily. Overall, the level of coupling coordination is low, and the type of coupling coordination has gradually evolved from severe and moderate imbalance to moderate coordination and mild imbalance. Therefore, the CCCG should make full use of the advantages of the “dual-core” cities to improve the level of peripheral economic linkage, increase investment in science and technology to enhance the endogenous power of economic development, establish cooperation mechanisms to narrow the urban gap, and integrate ecological resources to promote ecological industrialization to better realize the synergistic promotion of land ecological protection and high-quality economic development.

## 1. Introduction

Land ecosystem services are the benefits that humans derive from land ecosystems, which are an important material basis for sustaining people’s livelihood capital and livelihood activities and a key factor in enhancing people’s livelihoods [1,2]. Economic–social development is the basic guarantee for the construction of a sustainable China, and the construction of a sustainable China requires continuous resource input. As the carrier of all social and economic activities carried out by human beings, land plays an important role in the process of economic–social development [3,4]. In recent years, with the rapid economic and social development and the significant increase in population, the land use structure in China has shown a dramatic increase in construction land as well as a large occupation of agricultural land, especially arable land, and the land ecosystem services have been unable to meet the growing human demand for ecological environment. The excessive claim and use of ecological resources by urban construction not only cannot maintain their livelihood levels, but also exacerbates the deterioration of the land ecosystem and causes damage to the ecological environment that is difficult to recover [5]. Only by finding the “balance point” between urban economic–social development and land ecosystem can we implement the concept of regional ecological and economic development, promote the healthy development of urban economy, and truly achieve the region’s sustainable development [6,7]. Therefore, how to reconcile the contradiction between land ecosystem and economic–social development is the key to the current land use. Meanwhile, the Future Earth Plan, the UN <2030 Agenda for Sustainable Development>, and the Habitat III Conference emphasize that the urbanization process should develop in harmony with the ecological environment and adapt to the carrying capacity of resources and environment, and the key is how to coordinate the relationship between human and land [8,9,10]. How to consider economic development and land ecosystem as an entity and promote the development of the entire system in a positive, healthy, and mutually reinforcing direction on the basis of reasonable planning is an urgent problem to be solved at present.

The relationship between economic–social development and land ecosystems is complex, and a large number of studies have been carried out at home and abroad from individual perspectives. One is to determine the value of land ecosystem services by continuously improving the value equivalent factor based on the value scale developed by Costanza and others [11]. The other is the “equivalent factor method”, which is based on the “value of land ecosystem services per unit area of different terrestrial ecosystems in China” established by Xie Gaodi and others. According to the division of land use types, the value of land ecosystem services is assessed for different land use types [12]. The equivalent factor method is intuitive and easy to use compared with other valuation methods. It has been widely used and can estimate the value of land ecosystem services more effectively at large scales [13,14]. Studies on economic–social development have mainly used mathematical statistics and models based on a sustainable development framework to conduct in-depth studies on the coordination of land use and economic–social development. The methods used include correlation analysis [15,16,17], analytic hierarchy process [18,19], grey correlation method [20,21,22], and coupling degree function method [23,24]. Most scholars on the research content subdivide land use and economic–social system into several index layers, such as land use degree [25,26], industrial structure [27,28], urbanization level [3,29], and economic power development [27,30,31] to conduct research.

In general, although some scholars have analyzed the relationship between economic development and land use, the coupling and coordination relationship between economic–social development and land ecosystem services has been less examined. At the same time, the current studies on economic systems and ecosystem linkages are mostly based on single statistics, focusing more on the relationship between urbanization and ecological environment, and the value of ecosystem services that characterize the level of sustainable development of city groups lacks attention, as well as the integrated and quantified results of regional ecological environment changes. More importantly, the spatial scales of previous studies on the coupling coordination between land use and social–economic development have mainly focused on the national [32,33], economic belt [34], regional [35,36], provincial [37,38,39], and individual cities [40,41], while there are fewer studies related to the coordination between economy and environment at the scale of city groups. A city group is a relatively complete urban “aggregate” formed by multiple cities attracting and clustering with each other. As the highest form of regional spatial organization, city groups have become the core growth pole of China’s economy in the future [42]. Among them, the Chengdu–Chongqing city group (CCCG) is the fourth largest city group in mainland China after the Yangtze River Delta, the Pearl River Delta, and the Beijing–Tianjin–Hebei city group. It is also an important platform for the development of western China, a strategic support for the Yangtze River Economic Belt, an important demonstration area for urbanization, and an important economic center in China. Since 2011, the Chinese government has introduced a number of policies to promote the development of the CCCG, making it an economic cluster alongside the Beijing–Tianjin–Hebei city group, the Yangtze River Delta, and the Guangdong–Hong Kong–Macao city group, which has a profound impact on the development of China’s inland opening strategy [43,44]. However, the problem of resource wastage and environmental degradation in the CCCG is becoming increasingly prominent, and it has become one of the regions with the most obvious human–land conflicts. The land ecosystem and economic–social development system can reflect the complex human–land relationship. The economic–social system and land ecosystem are the two systems most closely linked to human survival and development. The relationship between ecosystem services and economic–social development is a non-linear, open dynamic change system, and the two are mutually influencing, forming an integrated system that can realize the adjustment of system structure and the optimization of function through coupled and coordinated research [45,46]. The coupled and coordinated relationship between the two can be summarized as shown in Figure 1. Land ecosystem services are the environmental conditions and utilities formed by the ecosystem to maintain human survival and development, mainly including supply services, regulation services, support services, and cultural services, which constitute the material basis of economic–social development. Economic–social development is the consumption carrier of land ecosystem services. Human economic–social development activities and lifestyle changes will strengthen the degree of human demand on the ecosystem, and if the demand for ecosystem services in the process of economic–social development exceeds its limits, it will certainly be constrained by the ecosystem. When economic–social development reaches a certain level, sustainable policies, which are favorable for the local economic and ecological conditions, can be introduced to promote coordinated and high-quality development. Therefore, it is an inevitable trend to strengthen the coordinated ecological and economic development of urban agglomerations, and it is also the core of research on sustainable development [42].

Based on this, this study analyzes the coupling coordination degree model of land-use remote sensing data and socioeconomic development data within 2005–2020 from the scale of city group using the entropy weight method and equivalent factor method, and carries out empirical analysis through the method of central migration and standard deviation ellipse, with a view to: (1) assessing the overall situation of land ecosystem services and economic–social development in the CCCG; (2) analyzing the change process of land ecosystem services and economic–social development in the region over a long period of time; and (3) analyzing the coupling, coordination degree between land ecosystem services and economic–social development in the region and the characteristics of spatial and temporal evolution, so as to provide a reference for the CCCG or other city groups in analyzing the coordinated development of land ecosystem services and economic–social development.

## 2. Materials and Methods

### 2.1. Study Area Overview

The CCCG is located in southwest China (27°39′–33°03′ N, 101°56′–110°11′ E), in the upper reaches of the Yangtze River and in the hinterland of the Sichuan Basin. It is also located at the intersection of the horizontal axis of the Yangtze River corridor and the vertical axis of the Baokun corridor in China’s “two horizontal and three vertical” urbanization strategy, which includes 16 cities, such as Chongqing, Chengdu, Zigong, Luzhou, Deyang, Mianyang, Suining, Neijiang, Leshan, Nanchong, Meishan, Yibin, Guang’an, Dazhou, Ya’an, and Ziyang, with a total area of about 185,000 km^2^ (Figure 2). The topography of the region is low in the west and high in the east, with a complex and diverse landscape. The CCCG is mainly plains and hills, with the areas to the east being hilly areas with complex and different landscapes and the areas to the west being the western Sichuan plains with a flat topography. The CCCG has a subtropical monsoon climate, with abundant precipitation yearly. At the same time, rain and heat are produced, bringing an average annual precipitation of 1000–1300 mm. Relying on its superior location advantage, resource endowment, industrial base and human resources, and centered on Chongqing and Chengdu, the CCCG is an important platform for western development, a strategic support for the Yangtze River Economic Belt, and an important demonstration area for the national promotion of new-type urbanization. In recent years, the CCCG has faced multiple problems, such as unbalanced regional development, the disorderly spread of urban expansion, and low land-use efficiency [47], which bring challenges to the sustainable development of the region, in parallel with the rapid social and economic development.

### 2.2. Data Source and Processing

The economic–social development data required for this study were obtained from <The Sichuan Statistical Yearbook>, <Chongqing Statistical Yearbook>, and <China City Statistical Yearbook>, and individual missing data were obtained from the differential complements. The land ecosystem data of the CCCG for 2005, 2010, 2015, and 2020 were obtained from the Resource and Environment Science and Data Center: https://www.resdc.cn/ (accessed on 3 June 2022), with a resolution of 1 km. The data are based on the Landsat satellite images and generated by manual visual interpretation, including 6 primary classes and 25 secondary classes of arable land, forest land, grassland, water, unused land, and urban–rural, industrial and mining, and residential land, while the CCCG covers 23 secondary classes out of 6 primary classes (Figure 3). The data processing and analysis were mainly conducted in the ArcGIS 10.2 software platform.

### 2.3. Research Methodology

#### 2.3.1. Assessment of the Level of Economic–Social Development

Drawing on the results of previous publications, this study identified suitable evaluation indicators from the perspectives of economic strength, economic structure, economic vitality, social foundation, and social services, and establishes a set of evaluation systems that can reflect the level of social development (Table 1). Social and economic strength is the overall performance of the level of regional economic development, and the per capita GDP was selected as the evaluation index in this paper [48,49]. Economic structure is a reflection of regional development structure and advanced, and this study used the proportion of tertiary industry in GDP and Education Expenditure to characterize it [50,51]. Economic dynamism represents the potential for sustainable development and development capacity of the regional economy, and is characterized by retail sales of social consumer hoods (RCSH) [52,53]. Social base is the concentration of regional population and the development of towns and cities, characterized by population density and final number of employees in urban units [54,55]. Social services are the reflection of the development level of human life in the region, and this paper chose education development (the ratio of primary and secondary school students), employment development (characterized by the number of industrial enterprises above designated size), and the level of medical development (characterized by the number of hospital beds per 10,000 people) [56,57,58]. In order to unify the magnitudes of the data of different indicators, this study normalized the raw data by extreme values.

The specific process is as follows:(1)$$x_{ij}^{\prime }=\frac{x_{ij}-x_{j\min}}{x_{j\max}-x_{j\min}}$$
where $x_{ij}^{\prime }$ is the standardized value of indicator j of city i; $x_{ij}$ is the original value of indicator j for city i; $x_{j\max}$ is the maximum value for indicator j; and $x_{i\min}$ is the minimum value for indicator j.

The weight of each index needs to be reflected by the entropy weight method (Table 1). The formulas involved are as follows:

Calculate the proportion of the index value of indicator j for city i:(2)$$y_{ij}=\frac{x_{ij}^{\prime }}{\sum x_{ij}^{\prime }}$$

The information entropy value of indicator j:(3)$$e_{j}=-k\sum_{i=1}^{m}y_{ij}\ln y_{ij}$$
where k=1ln(m), k > 0.

The information entropy redundancy of indicator j:(4)$$g_{j}=1-e_{j}$$

The weight of indicator j:(5)$$w_{j}=g_{j}/\sum_{j-1}^{n}g_{j}$$

The overall development level of the economic–social system can be obtained by the linear weighting method:(6)$$U=\sum_{i-1}^{n}w_{j}\times x_{ij}^{\prime }$$
where U is the overall level of regional economic–social development; $w_{j}$ is the weight of indicator j in the economic–social system; and $x_{ij}^{\prime }$ is the standardized value of indicator j of city i.

#### 2.3.2. Assessment of the Level of Economic–Social Development

This study drew on the equivalent factor method proposed by Xie Gaodi to calculate the land ecosystem services [59] and modified it. The correction process is as follows:(7)$$e=\frac{1}{7}\times (P\times Y)$$
(8)E=e×q
(9)$$ESV=\sum_{j=1}^{m}\sum_{i=1}^{n}A_{i}E_{ij}$$
(10)$$AESV=\frac{\sum_{j=1}^{m}\sum_{i=1}^{n}A_{i}E_{ij}}{\sum_{i=1}^{n}A_{i}}$$

In Equations (7)–(10), e represents the economic value of food produced by the ecological economy of farmland per unit area; P is the average price of food; Y is the yield per unit area of food; q represents the land ecosystem service value equivalent per unit area; E is the ecosystem service value per unit area; ESV is the land ecosystem service value; A_i_ is the area of land type i in the ecosystem. E_ij_ is the j-th type of ecosystem service value of the i-th type of land within unit area; and AESV is the land ecosystem service value of land on average.

As shown in Table 2, the land ecosystem service values per unit area of CCCG were obtained according to the correction method. In terms of land types, the ecological service value of watershed accounts for 67.10% of the total land ecosystem service value; the ecological service value of forest land accounts for 18.85% of the total land ecosystem service value, followed by grassland and farmland with 7.82% and 5.29%, respectively; and desert accounts for only 0.93%. In terms of ecosystem service types, hydrological regulation has the highest service value, accounting for 25.91% of the total value, followed by waste treatment and climate regulation, accounting for 22.75% and 14.97% of the total value, respectively; and food production and raw material production have the lowest service value, accounting for only 1.79% and 2.92% of the total value, respectively.

#### 2.3.3. Coupling Coordination Degree Model and Calculation

The degree of coupling measures the strength of the interaction between the system or elements, and can provide an early warning of the development order [60]. The coupling coordination degree model of this study was as follows:(11)$$D=\sqrt{C\cdot T}$$
(12)$$C={\left\{\frac{E_{1}\times E_{2}}{{\left[(E_{1}+E_{2})/2\right]}^{2}}\right\}}^{1/2}$$
(13)$$T=a\cdot E_{1}+b\cdot E_{2}$$

In Equations (11)–(13), D represents coupling coordination degree (0 ≤ D ≤ 1). The larger the value, the more coordinated the development of the two systems. C represents the coupling degree between land ecosystem services and economic–social development (0 ≤ C ≤ 1). The smaller the value, the worse the coupling degree between the two systems and the trend of disorder will be developed. E1 represents the comprehensive evaluation index of land ecosystem services; E2 stands for comprehensive evaluation index of social and economic development; T represents the comprehensive coordination index of the two systems; a represents the contribution rate of land ecosystem services to the system, let a = 0.5; and b represents the contribution rate of social and economic development to the system, let b = 0.5. The distribution function proposed by Liao Chongbin was used for the classification of coupling coordination degree (Table 3) [61].

#### 2.3.4. The Standard Deviation Ellipse

The standard deviation ellipse (SDE) is most commonly used statistical method for the evolution of time–space with the center of gravity, the major axis, the minor axis, and the rotation angle as parameters, which quantitatively analyzes the characteristics of the spatial analysis of the object [62]. The center of gravity is equivalent to the spatial distribution position of geographical elements, the azimuth is equivalent to the trend direction of the distribution of geographical elements, the major axis is equivalent to the dispersion degree of geographical elements in the main direction, and the minor axis is equivalent to the dispersion degree of geographical elements in the secondary direction.

## 3. Result and Analysis

### 3.1. Performance Level Trend Analysis

#### 3.1.1. General Analysis of Land Ecosystem Services and the Economic–Social Development

As shown in Figure 4, the overall level of economic–social development system is low, but shows an upward trend year by year. The index range is 0.088–0.276; this indicates that the urbanization and industrialization of the CCCG accelerated during this period, and the level of economic–social development increased significantly. At the same time, the impact of human activities on the ecosystem intensified, which is consistent with the result that the ecological service value decreased. The overall level of ecological service value is high, which is mainly due to the steady promotion of the national “return of cultivated land to forest” project, and the vegetation restoration achieved some results; the index range is 0.703–0.720, with a slightly decreasing but stable trend from 2015. The ecological service value index of CCCG is always higher than the economic–social development index, and the gap between the two values is constantly narrowing. This indicates that the ecological foundation of the region is good and the situation of lagging development has been improved.

#### 3.1.2. Changes in the Service Value of Each Land Type in the Ecosystem Service

As can be seen from Table 4, among the various types of ecosystems, the service value of the forest land is the largest, with an average contribution rate of 58.14%. The contribution rate of the cultivated land is located in the second place, with an average contribution rate of 26.55%, and the ecological value also shows a decreasing trend. The cultivated land plays a very important role in the entire ecological service system, not only in relation to food security, but also in relation to the climate and waste disposal of the region and many other aspects [63,64]. The contribution of grassland and watershed are in the third and fourth places, respectively, and the ecological service value of grassland shows a decreasing trend, while that of water area shows an increasing trend. Both are an integral part of the service value system. The water area system is closely related to production and waste disposal, and the decreasing trend of these factors cannot be ignored [65].

#### 3.1.3. Changes in the Development of the Factors of Economic–Social Development

In the economic system, the average rate of change in education expenditure is the largest, which is 84.32%, with an overall plummeting upward trend (Table 5). Education investment is a basic and strategic investment to support the long-term development of the country, an important cornerstone of national development and social progress, and an important part of the economic–social development system [50,51]. The average rates of change in retail sales of social consumer goods and per capita gross regional product are in the second and third places, with average rates of change of 49.71% and 44.49%, respectively. Both are related to people’s productivity and consumption level, and provide indirect feedback on people’s happiness index, which plays a very important role in the entire economic–social development system [52,53]. Both also show a steady upward trend, but the rate of change in population density (−3.82%) shows a downward trend of change, although it is the smallest. The change in population density is a double-edged sword; its growth can promote local economic development but it can bring serious urban congestion, environmental damage, and other problems [54,55]. The number of hospital beds occupied per 10,000 people, the number of urban workplaces employees at the end of the period, the number of industrial enterprises above the scale, and other factors are closely related to economic–social development, and the upward trend of these factors is crucial. The specific results of the development of each factor of economic–social development are as follows.

### 3.2. Economic-Social Development and Changes in the Value of Land Ecosystem Services

#### 3.2.1. Changes in the Level of Economic–Social Development

In terms of the economic–social development level index, Chongqing and Chengdu in CCCG have extremely high and prominent levels of economic development, with extremely obvious differences with other cities, while the differences between other cities are smaller and at low levels (Table 6 and Figure 5). During 2005–2020, the development gap between Chongqing and Chengdu and other cities increased, despite the steady upward trend of the economic–social development of each city. The possible reasons for this are that Chongqing and Chengdu are both important in China’s urban development, with good geographical and social factors; they also have the advantage of unique tourism resources and increasing quality talent gathering, which makes the social and economic level of these two cities considerably high. From the ranking of economic–social development level in the period of 2005–2020, Chongqing and Chengdu rank first and second, respectively, in the whole CCCG; Deyang and Mianyang have similar economic–social development rankings, showing a trend of ladder-like changes. The economic–social development rankings of Neijiang and Zigong show a plummeting trend; the economic–social development rankings of Nanchong show a “backward ladder” trend; and the economic–social development rankings of Suining, Ziyang, and Leshan show an overall rising trend followed by a falling trend. The other cities are relatively lagging in overall ranking, but also show an upward trend. These areas are relatively remote in location, backwards in transportation and cultural resources, and have a relatively serious population loss; therefore, their economic base is relatively weak [37,38,39]. In general, since 2010, the regional development planning of CCCG has increased, and the development of each region has sped up. Relying on their natural advantages and supplemented by various national urban development incentive policies, the cities have developed rapidly in terms of economic–social development.

#### 3.2.2. Variation in the Value of Land Ecosystem Services

From Table 7 and Figure 6, it can be seen that the land ecosystem service value of each city in CCCG is relatively obvious, among which Chongqing and Mianyang have the highest overall land ecosystem service value, and the land ecosystem service value of Chongqing is significantly different from that of other cities, followed by that of Dazhou and Ya’an, with Zigong and Ziyang being relatively low. The low value of the land ecosystem service is mainly related to the local population migration and the low human modification of the land surface [66,67,68]. Overall, the variability characteristics of the land ecosystem service values vary with relatively large value differences.

The introduced average land ecosystem service value (AESV) can characterize the ecological service differences among different areas of the CCCG. The results of this study show that the mean value of the average land ecosystem service value index decreased from 29,999 in 2005 to 29,840 in 2020. Among them, the AESV was the largest in both Leshan and Ya’an cities. These areas have more deep forest cover, better resource protection, and stronger ecological service function of watershed and woodland. In contrast, although Ziyang City and Neijiang City have fewer human activities and less disturbance of land ecosystem services, there is more exploitation of land resources, so the overall ecological service value is low [69,70]. In conclusion, the overall urban ecological service value of the CCCG is higher.

### 3.3. Coupling Coordination Analysis between Economic–Social Development and Ecological Service Values in CCCG

#### Analysis of the Evolution of the Coupling Degree between Economic–Social Development and Ecological Service Value

As can be seen from Table 8, the overall coupling degree of the CCCG increased from 0.50 in 2005 to 0.66 in 2020, and the coupling coordination level changed from 5 to 4, with a slow growth trend in between, and the overall coupling degree of ecological service value and economic–social development increased.

During 2005–2020, the development of the coupling coordination degree of ecological service value and economic–social development of each city in the CCCG has a more obvious level difference (Table 9 and Figure 7). Overall, the level of coupling coordination among cities in the CCCG is low and shows a continuous and steady increasing trend. Among them, the coupling coordination degree of Chongqing is high and changed the most, and the economic-social development and land ecosystem services of Chongqing in 2020 are at a high coupling stage with strong mutual coupling. The second one is Chengdu, at a medium coupling stage. Ziyang, Zigong, Suining, Neijiang, Deyang, Guang’an, and Ya’an are at a low coupling stage with a low coupling coordination. Since 2010, the coupling degree of the CCCG and the overall coordination degree has improved more obviously, but most cities are still at a stage of mild disorder. From the entire process of change, although the coupling degree of all regions in the CCCG has been greatly improved, the characteristic of “strong core and weak periphery” is more prominent. In recent years, Chongqing achieved a high-quality coordination level, and Chengdu achieved a barely coordinated level, while many places, such as Yibin City, Dazhou City, Ya’an City, and Luzhou City, are still at a mild or below disorder level. Ziyang City, Suining City, Guang’an City, and Zigong City are at a moderate or below coordination level. To a certain extent, CCCG has not effectively resolved the contradiction between economic economic–social development and land ecosystem protection, the coordination between economic development and land ecological construction is not strong. The main reason is that Chengdu and Chongqing, as the capital of Sichuan province and municipality directly under the central government, respectively, are at the forefront of the gradient transfer of regional economic development by taking advantage of their political and economic location, and have a certain priority in development compared with other cities. After the economy of Chengdu–Chongqing cities reaches a certain scale, they will continue to look for economic depression areas, and other cities in CCCG will further undertake labor-intensive industries in Chengdu–Chongqing. The development of traditional industries, such as relying on undertaking industries with resource development, agricultural and sideline industries, and the chemical industry, has a spontaneous infringement mechanism on the ecological environment, so the CCCG has a low degree of coupling and coordination in the process of economic development to different degrees. In this regard, the “dual-core” cities focus on the development of a modern service industry, high-tech industry, and advanced manufacturing industry, and emphasize the development of the surrounding areas led by the headquarters economy. The planning of CCCG will further strengthen the interconnection of transportation and communication infrastructures, regional collaborative innovation, joint prevention and control of pollution in the region, thus enhancing the connection between the “dual-cores” and the surrounding cities, promoting the integrated development of ecology, transportation, industry, and market, and forming a city group with strong re-radiation capacity, close economic connection, and reasonable system structure.

### 3.4. Analysis of Spatial–Temporal Heterogeneity of CCCG

#### 3.4.1. Spatial and Temporal Patterns in the Value of Land Ecosystem Services on a Land-Average Basis

The spatial differences in the average land ecosystem service values of CCCG are significant (Figure 8). Overall, it shows a “three-core” high-value spatial cluster structure with Mianyang, “Ya’an–LeShan”, and “Chongqing–Luzhou-Dazhou” as the core, while the central, western, and northern parts of the city have a lower value. From 2005 to 2020, the spatial characteristics of each city did not change much, and regressive development of the value of land ecosystem services occurred only in a few cities.

#### 3.4.2. Spatial and Temporal Patterns of Economic–Social Development in CCCG

From 2005 to 2020, the economic–social development level of CCCG showed a slow upward trend (from 0.088 to 0.276), with a low overall level and obvious spatial differences (Figure 9). In 2005, the overall level of economic–social development was low with Chengdu and Chongqing as the core of the high-value area of economic–social development, and the level of economic–social development in the northern, central, southwestern, and southern cities was low. A “U”-shaped spatial structure was formed along Neijiang and Zigong. In 2010, the economic–social development level of the CCCG was generally reduced due to the influence of the general environment, such as the 2008 “5–12” earthquake and the international financial crisis, and the high-value and higher-value areas of economic–social development were distributed in a clan pattern with Chengdu and Chongqing as the core, while the low-value areas increased and were mainly distributed in the central, southern, northern, and Ya’an cities. In 2015, the spatial difference in regional economic–social development level decreased, the high-value area and higher-value area was distributed in the “Chengdu–De-Mian” urban belt and Chongqing city, and the low-value area was located in the central and western cities. In 2020, the spatial difference in regional economic–social development level decreased significantly, the high-value area appeared in the main urban areas of Chengdu (0.68) and Chongqing (0.78), the higher-value area increased significantly, and the low-value area had a grouped spatial distribution with Ziyang city as the core. The economic–social development level of CCCG show an “inverted U”-type spatial structure with the main urban areas of Chengdu and Chongqing as the core and along the “De-Mian–Nan-Da” urban belt.

#### 3.4.3. Spatial and Temporal Patterns of Economic–Social Development and Ecosystem Coupling in the CCCG

From the spatial evolution of the coupling coordination degree (Figure 10), the coupling degree of economic–social development and ecological service system in CCCG in 2010, 2010, and 2015 shows an obvious upward trend, and the cities within the CCCG gradually show a benign upward development. The system as a whole is well-developed, evolving from the periphery surrounding the central part and the north moving to the south. However, in 2020, the coupling degree of most cities in the CCCG gradually developed downward, with Chengdu and Chongqing as the only cities with a high coupling degree. Mianyang and Dazhou were the only cities with a high coupling degree, and a significant increase in the number of low coupling degree areas can be observed. Showing a concave spatial structure with “two prominent cores, a collapsed central part, and a higher periphery than the interior”, Chengdu and Chongqing have obvious advantages in economic development and ecological construction by virtue of their location and optimized industrial structure, and have crossed the potential barrier area to achieve a high level of coordinated regional economic and ecological development. By the end of the study period, only Chongqing reached the stage of high-quality coordination, and the coupling between the majority of cities needs to be further improved. The low value of coupling is mainly due to the large proportion of traditional industries characterized by high-input, low-output, and high-energy consumption and heavy pollution; the population potential is not fully stimulated; urban and rural infrastructure construction is generally lagging; and ecological deterioration and resource depletion are increasingly prominent. These cities should seize the opportunities of “The Belt and Road” construction, CCCG planning, and Chengdu–Chongqing industrial transfer to understand their comparative advantages in terms of economic location and be integrated into the national higher-level allocation of resources for industrial and economic transformation and upgrading. Each city should take into account its own development position and objectives, formulate corresponding development strategies, and adopt a series of comprehensive measures to enhance the synergistic development of the dual system of economy and environment in the region. They should also promote their own sustainable development and further contribute to a coordinated and high-quality regional development.

To further investigate the spatial evolution process of the coupling coordination degree of ecological service value and economic-social development in the CCCG, the standard deviation ellipse and the center of gravity migration trajectory of the overall coupling coordination degree of the CCCG from 2005 to 2020 were obtained using ArcGIS. The corresponding standard deviation ellipses were plotted according to the base standard deviation ellipses of the CCCG in each time period [22]. Our study found that, in 2005–2020, the center of gravity generally moved in the northeast direction with a distance of 3.2 km and then shifted to the southwest with a distance of 0.31 km, and the shift in the center of gravity in the north–south direction was slight (Figure 11). Analyzing the morphology of the ellipse (Table 10), the ellipse shifted eastward from 2005 to 2015 with a significant flattening rate. This flattening rate became larger, which indicates that the distribution pattern was relatively divergent, and the rotation angles were 76.88°, 77.71°, and 78.26°. From 2015 to 2020, the position of the ellipse was pulled back, and the difference between the long and short axes was significantly reduced with a decreased flattening rate. The changes in the area, axis length, and rotation angle of the standard deviation ellipse of the coupling coordination degree of ecological service value and economic development were relatively small during the study period, which indicates that the spatial structure of the coupling coordination degree of ecological service value and economic–social development in the CCCG is relatively stable. Overall, the degree of coupling and coordination of ecological service value and economic–social development in CCCG shifted toward Chongqing, indicating that the degree of coupling and coordination in Chongqing and the surrounding areas has improved greatly in recent years, and the pulling effect of cities relative to other reginal cities to the value of ecological service and economic-social coupling and coordination has been enhanced. The reason for this may be that the overall development status of Chengdu occurred earlier than that of Chongqing. Therefore, most early studies on the expansion of the western region focused on the periphery of Chengdu. As the twin-city economic circle strategy deepens and the socio-economy is well-developed, the momentum of outward extension to cities and towns will continue to weaken after the city reaches a certain level of development and the center of gravity begins to favor Chongqing city. From the direction of the center of gravity shift during 2015–2020, the territory of Chongqing city is an important area, and the expansion momentum of Chongqing area will be enhanced.

#### 3.4.4. Spatial and Temporal Evolution of the Type of Coupling between Economic–Social Development and Ecosystem Services in the CCCG

The coupling coordination degree of economic–social development and ecological service system of CCCG in 2005, 2010, 2015, and 2020 were selected for analysis, and the results are shown in Figure 12.

Overall, the coupling and coordination level of economic–social development and ecological service system of CCCG show a trend of steady improvement, but the overall coupling and coordination degree is poor, with the coupling and coordination degree decreasing from 16 to 14 cities in the disorder region from 2005–2020. There are more dysfunctional cities due to a lagging economic-social development, indicating that economic–social development is an important influencing factor for high-quality regional development. Analyzing at the level of cities, the spatial differences in the level of coupling and coordination between economic–social development and ecological service system in the CCCG are relatively obvious. Three cities in the Chengdu–Chongqing region were in serious disorder in 2005, and ten cities were in moderate disorder, accounting for most of the cities in the entire Chengdu–Chongqing region. These cities’ synchronization is hindered mainly because of the interaction of the poor regional economic environment and economic lag. Among them, Chengdu, as the provincial capital, and Chongqing, as the municipality directly under the central government, have lagged in ecological service level and economic development, respectively. In 2010, there was 1 region with severe disorder, 10 regions with moderate disorder, and 1 region on the verge of disorder, and the economic–social development and ecological service system in the entire economic circle were at a low level. Among them, Chongqing’s economic lag was the main reason for the imbalance in coordination level in the region. In 2015, there was one region with a serious disorder, six regions with moderate disorder, and seven regions with mild disorder. The level of coupling between economic–social development and ecological service system did not considerably change in cities such as Leshan and Mianyang, while the economic lag in the Chongqing region was more serious than that in 2010. In 2020, the coupling between the economic–social development and ecological service system of many cities increased, and Chongqing, as a city with a high level of coupling, had excellent policy welfare advantages and was an innovative pilot city with considerable national and good policy support. It has become a type of zone where social and economic development and ecological service system develop together [71]. From 2005 to 2020, Deyang, Zigong, Neijiang, Suining, and other cities in the economic circle, which are relatively lagging in economic development, developed greatly during the 13th Five-Year Plan period, and the level of economic development improved, while the “siphon effect” diverted a large amount of high-quality resources to Chengdu and Chongqing, resulting in the lagging development of science and technology in the region. The natural resource advantages of Chengdu and Chongqing and the strategy of “Chengdu–Chongqing dual-city economic circle” have led to the development of a barely coordinated and high-quality coordinated type. Chengdu, Yibin, Luzhou, and Nanchong experienced rapid economic development during the study period, but ecosystem protection lagged behind economic development, making them ecological laggards. This indicates that the rapid economic development of cities can damage the service function of natural ecosystems to a certain extent. CCCG has developed greatly with the attention of the national and local governments, but there is still a vast gap compared with the set goal of high-quality development, and further coordination between economy and ecology is needed to achieve coordinated, high-quality, and sustainable development.

## 4. Discussion

This paper studied the coupling and coordination relationship between land ecosystem services and economic–social development in the CCCG from 2005 to 2020, taking 16 prefecture-level cities in CCCG as its research object.

(1)The comprehensive land ecosystem service index of CCCG declined slightly, with obvious regional differentiation

Through the calculation results of the comprehensive level index of land ecosystem services in CCCG, it was concluded that the comprehensive level index fluctuated slightly, decreasing in the last 15 years from the perspective of time evolution (decreasing 0.017 in 2005–2020). According to the calculated data, it can be divided into three development stages: the first stage (2005–2010) is the exploratory stage, in which the index decreases but at a relatively mild rate; the second stage (2010–2015) is the growth stage, in which the index increases; and the third stage (2015–2020) is the post-fall stage, during which the rate of decrease is high. This may be due to the rapid urbanization and the replacement of the original natural surfaces by urban construction sites, resulting in bare ground, many man-made buildings, and sparse vegetation, thus negatively affecting the urban ecological environment, which is consistent with previous studies [72,73]. It is because of the far-reaching impact of human activities on the structure and spatial changes in regional ecosystems and the increasingly serious problem of global ecological degradation that sustainable development has become one of the most important global themes [74,75]. In terms of spatial evolution, urban development is uneven and there are serious differences. Spatial distribution was also observed to have a convergence effect with the results of the integrated level index of land ecosystem service value. The formation of spatial differences in urban land ecosystem services may be influenced by receiving local government policies and ecological restoration projects [76,77]. Therefore, in order to protect the land ecosystem services in the future, we should follow the principle of sustainable development and try to effectively solve the existing ecological problems and create good conditions for economic–social development.

(2)Linear growth of the comprehensive level index of economic–social development in CCCG with unbalanced regional development

Through the calculation results of the comprehensive economic–social development index of CCCG, it was concluded that, from the perspective of time evolution, the comprehensive economic–social development index of CCCG showed a linear upward trend in the past 15 years (0.188 growth in 2005–2020), but the growth rate varied widely. According to the calculated data, it can be divided into three development stages: the first stage (2005–2010) has the slowest growth rate, the second stage (2010–2015) has a moderate growth rate, and the third stage (2015–2020) has the fastest growth rate. In terms of spatial evolution, the development of local cities was uneven and there were great spatial differences. Chongqing and Chengdu are at a high level, while Ziyang and Ya’an are at a lower level. There is a vast gap between the two. In terms of spatial location, the cluster effect is more obvious. The index is higher in the eastern cities and lower in the western cities. The reason for this is that, since 2000, China has proposed a western plan, which has increased support for the western region, and the establishment of various economic zones and high-tech zones within cities has accelerated the process of economic–social development [78]. However, influenced by the long tradition of unipolar development in Chengdu or Chongqing [79], there are obvious differences in the level of economic–social development in CCCG, resulting in a growing gap in economic–social development between other second-, third-, and fourth-tier cities in CCCG and the first-tier cities of Chengdu or Chongqing. However, in recent years, China has issued the “Chengdu–Chongqing City Cluster Development Plan”, which clearly requires CCCG to complete the historic leap from a national city group to a world-class city group by 2030. CCCG has gradually formed a “multi-city growth pole” situation, and the regional economic–social development gap has narrowed.

(3)Economic–social development and land ecosystem services are not synchronized and affect each other

According to the calculation results of the economic–social development and land ecosystem services, the integrated level of land ecosystem services is higher than economic–social development in each year of the study period, which indicates that land ecosystem services are more protected than the development rate of economic–social development. From 2005 to 2010, the land ecosystem services index showed a decreasing trend, while the economic–social development index increased at a modest rate. 2010–2015, the economic–social development index and the land ecosystem services index showed an increasing trend. From 2015–2019, the land ecosystem services index plummeted and the economic–social development index increased rapidly. It can be seen that economic–social development and land ecosystem services are not independent, but promote and limit each other. Economic–social development limits the improvement of regional land ecosystem services, and land ecosystem services also pose some constraints on economic–social development, which again verifies the environmental Kuznets inverted U curve, and the results of this study are highly consistent with those of previous studies [80,81]. It is important to focus on balancing the development of the two systems to make them stable and synchronous in the future.

(4)Low level of coordination between economic-social development and land ecosystem services

The calculation results of the coupling coordination degree show that the economic–social development of CCCG and the land ecosystem services are at the antagonistic stage, and the level of coupling coordination between them is low. In terms of temporal evolution, the overall coupling coordination degree of CCCG fluctuated in the range of 0.2–0.5 in the past 15 years, with a small interval, and was in an unbalanced state most of the time. From the perspective of spatial evolution, the coupling degree of prefecture-level cities was not very different; the coupling coordination level of Chongqing and Chengdu was relatively high, while that of Ziyang and Zigong was relatively low, and the differentiation between regions is obvious. In 2005–2020, the overall coordination level of CCCG developed in a good trend, and the gap between the coordination levels of prefecture-level cities narrowed, but there is still large room for optimization. The relatively poor level of the coupling and coordination of CCCG is strongly related to the geographical location of the Chengdu–Chongqing region, and the traditional rough development mode has a great negative impact on the coordinated development of urban economy and ecosystem. In the process of gradually approaching the strategic requirements, it is necessary to accelerate the industrial transformation, abandon the crude development mode of relying on nature, and develop tourism and technology industries with their own advantages as the starting point [82,83]. While ensuring the speed of development, it is necessary to take the improvement of environmental quality as a rigid indicator to avoid the problems caused by the single pursuit of development speed and further promote the benign development of economic–social development and land ecosystem services.

## 5. Conclusions

In this study, CCCG was taken as the research area, and the land-use remote sensing monitoring data of the city group in 2005, 2010, 2015, and 2020 were used to calculate the ecosystem service value in each period. Combined with the equivalent factor method, a comprehensive evaluation index system of the economic–social development of the city group was established, and the weight of each index was determined by entropy weight method. The economic–social development level of the city group was calculated by the linear weighting method. Finally, the coupling coordination degree model and standard deviation ellipse were used to quantitatively study the coupling coordination process between land ecosystem services and economic–social development in CCCG from 2005 to 2020. By analyzing the coupling coordination relationship between economic–social development and land ecosystem service of CCCG and its spatial and temporal evolution characteristics, the following main conclusions were obtained.

(1)From 2005 to 2020, the economic–social development of CCCG was generally on an upward trend, showing a “dual-core” spatial structure that was high in the east–west, low at the center, and the main urban areas of Chengdu and Chongqing as the core; the land ecosystem services as an entirety showed a gentle slope downward trend, and the entirety showed a “U”-shaped spatial pattern that was high around and low in the middle.(2)The economic–social development and land ecosystem service coupling coordination degree of CCCG continued to rise steadily, showing a spatial pattern of high around and low in the middle, and the overall coupling coordination level was low. The type of coupling coordination gradually evolved from severe and moderate imbalance to moderate and mild imbalance, and the degree of coupling coordination in most regions increased in an “upward” manner.

The research results show that most cities in CCCG still have much room for improvement in land ecosystem and economic–social development. In this regard, this study proposes the following suggestions:(1)To address the problem of regional differentiation of land ecosystem service levels, each city in CCCG should reasonably control the scope of construction land, improve the ecological protection compensation and regional joint management mechanism, strengthen infrastructure construction, abandon the crude development mode of relying on natural resources, speed up industrial transformation, improve the transportation and information networks, and develop ecological tourism and high-tech industries by highlighting their advantages.(2)In response to the overall low level of economic–social development, the government should adhere to the concept of ecological protection, using the advantages of each place, and develop ecological industries, whilst at the same time, rely on the location advantages of neighboring developed cities to actively introduce high-quality talents and high-tech industries to further improve their economic-social development. These cities should seize the opportunities of “The Belt and Road” construction, CCCG planning, and Chengdu–Chongqing industrial transfer to understand their comparative advantages in terms of economic location and integrate into the national higher-level allocation of resources for industrial and economic transformation and upgrading.(3)To address the overall low level of coupling coordination; to protect the ecological system while promoting the steady improvement of economic–social levels and achieving the coordinated development of regions; and to give full play to the “dual-core” city’s driving role, the “dual-core” cities should focus on the development of modern service industry, high-tech industry, and advanced manufacturing industry, and emphasize the development of the surrounding areas led by the economy of the headquarters. The planning of CCCG will further strengthen the interconnection of transportation and communication infrastructures, regional collaborative innovation, joint prevention and control of pollution in the region, thus enhancing the connection between the “dual-cores” and the surrounding cities, promoting the integrated development of ecology, transportation, industry, and market, forming a city group with strong re-radiation capacity, close economic connection, and reasonable system structure to promote coordinated regional development.

Although this study analyzed various elements of the coupled coordination level of economic–social development and land ecosystem services in CCCG in a comprehensive manner, there are still certain limitations. First, the construction land factor was not considered by the study. Second, the relationship between economic–social development and land ecosystem services is an open and complex giant system involving many elements, and other elements were not considered in this study. Further discussion is needed in future studies. Therefore, considering the universality of the coupled assessment model constructed in this study, when applying it to other regional studies, the research methods and models mentioned in this study can continue to be used, but the variability of ecosystem service value coefficients and the comprehensiveness of economic–social development indicators in different regions should be taken into account, and the value of construction land should be added. Based on this, it can be better applied to other regional studies.

## Figures and Tables

**Figure 1 ijerph-20-05095-f001:**
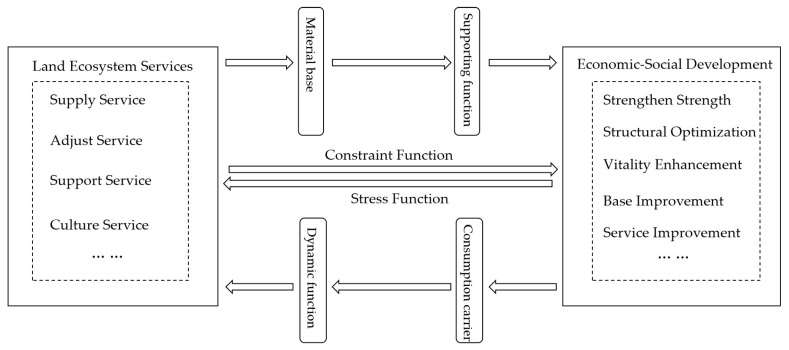
Relationship of the coupling coordination between land ecosystem services and economic–social development.

**Figure 2 ijerph-20-05095-f002:**
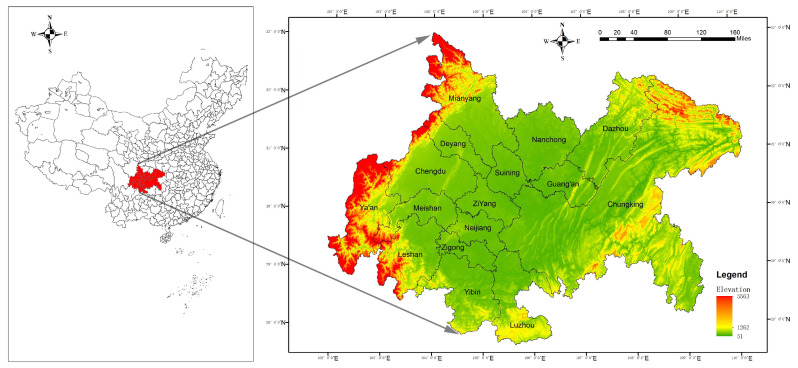
Geographical location of the Chengdu–Chongqing region and composition of the city groups.

**Figure 3 ijerph-20-05095-f003:**
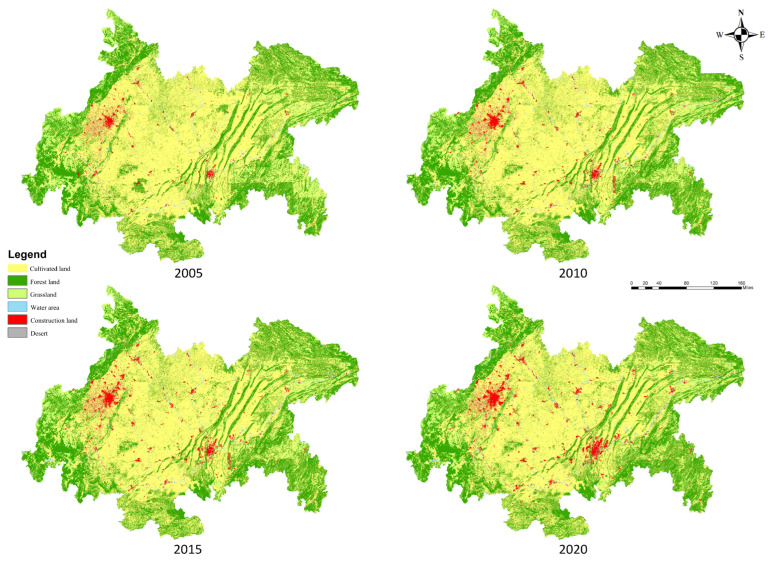
Ecosystem types of CCCG, 2005–2020.

**Figure 4 ijerph-20-05095-f004:**
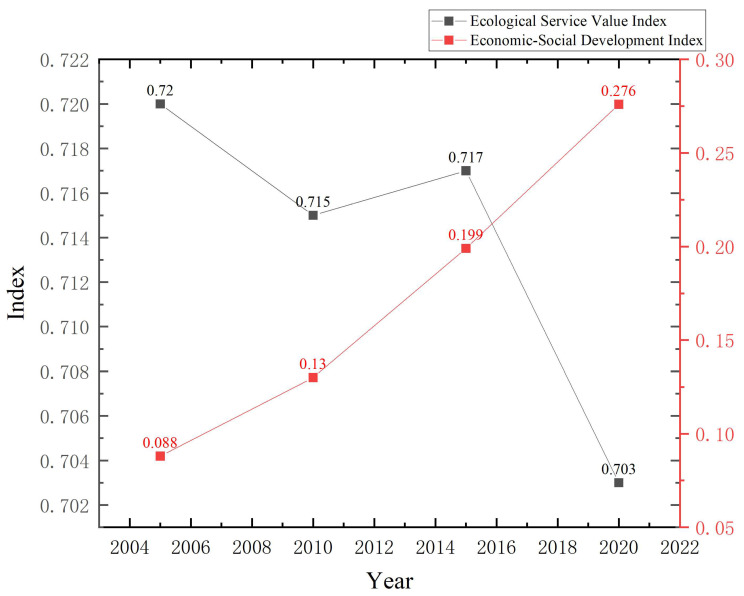
Time series of land ecosystem services and economic–social development indices in CCCG.

**Figure 5 ijerph-20-05095-f005:**
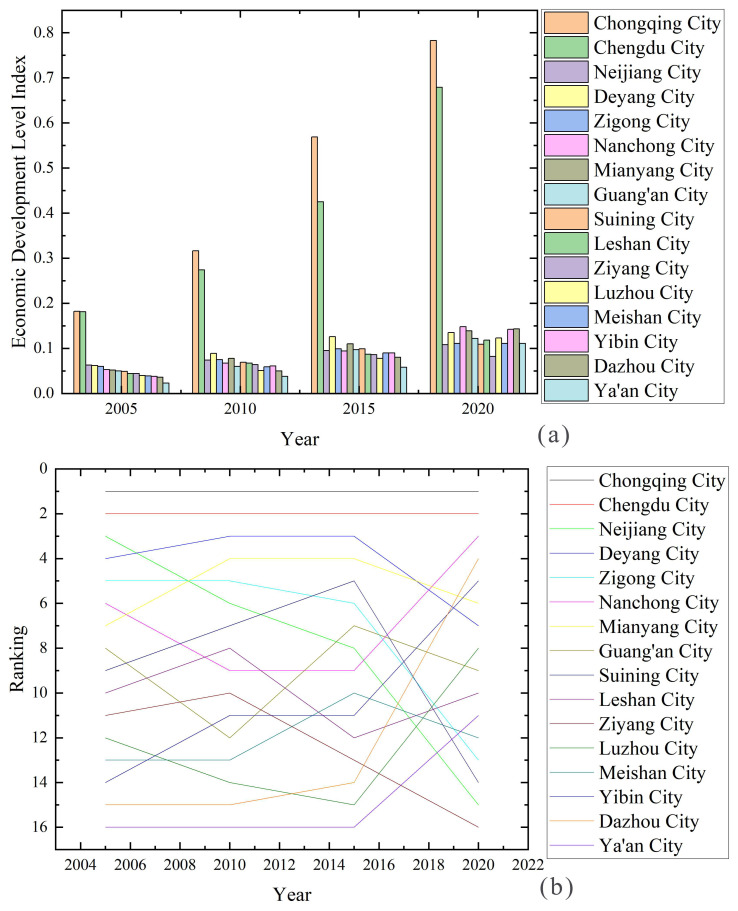
Economic–social development index and ranking of cities in CCCG, 2005–2020. (**a**) is a graph comparing the scores of cities. (**b**) is a graph comparing the ranking of cities.

**Figure 6 ijerph-20-05095-f006:**
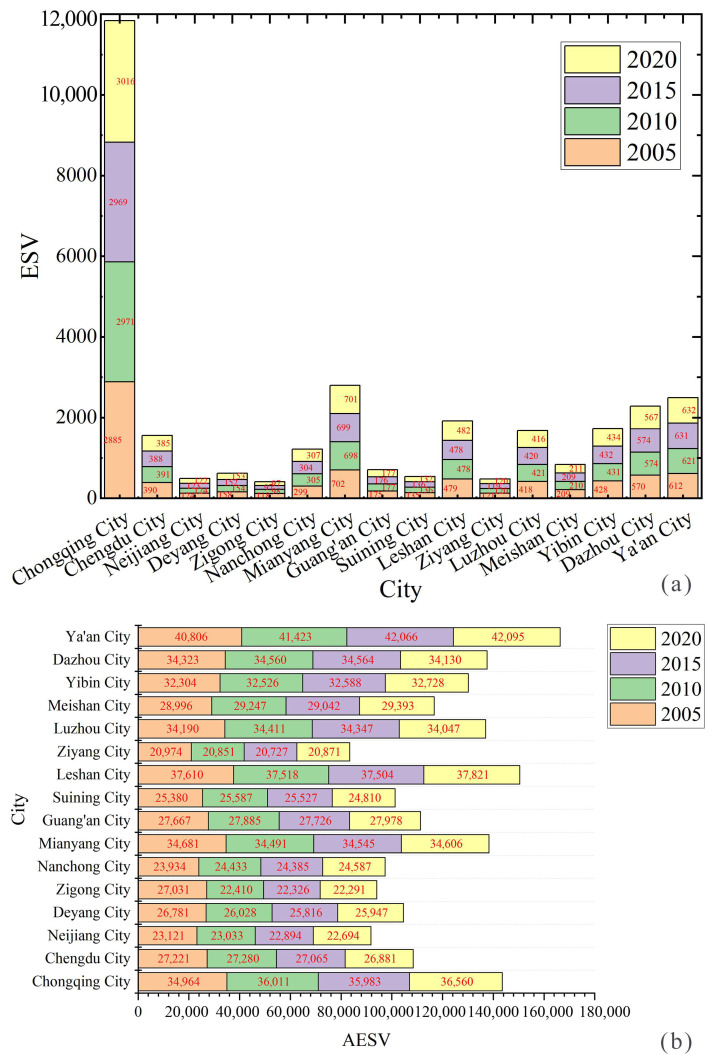
Changes in land ecosystem service values and land-average land ecosystem service values, 2005–2020. (**a**) is a graph comparing the ESV of cities. (**b**) is a graph comparing the AESV of cities.

**Figure 7 ijerph-20-05095-f007:**
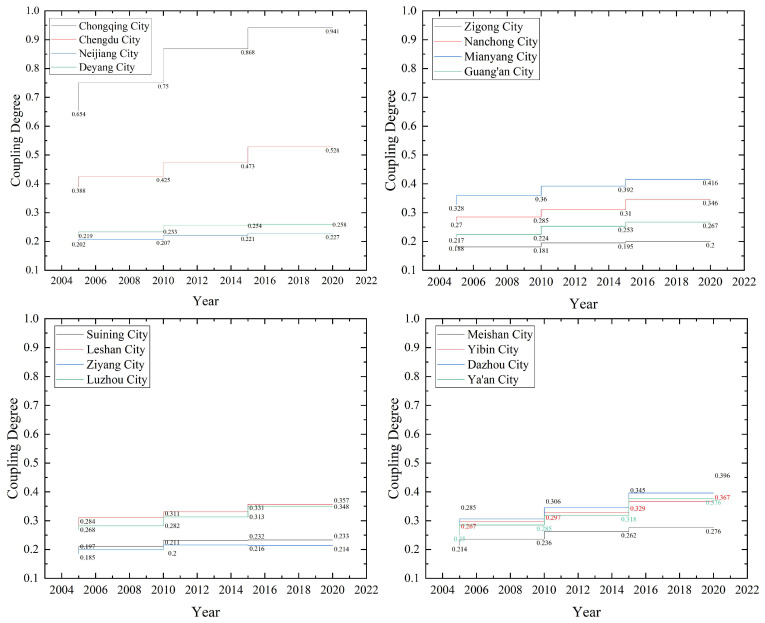
Trend of coupling and coordinated changes in economic–social development and ecological service system in CCCG from 2005 to 2020.

**Figure 8 ijerph-20-05095-f008:**
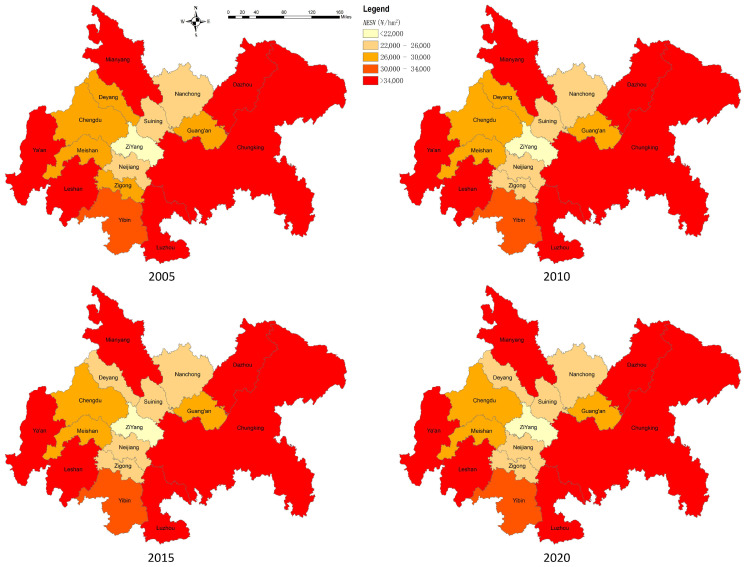
Change in land ecosystem service value per city in CCCG, 2005–2020.

**Figure 9 ijerph-20-05095-f009:**
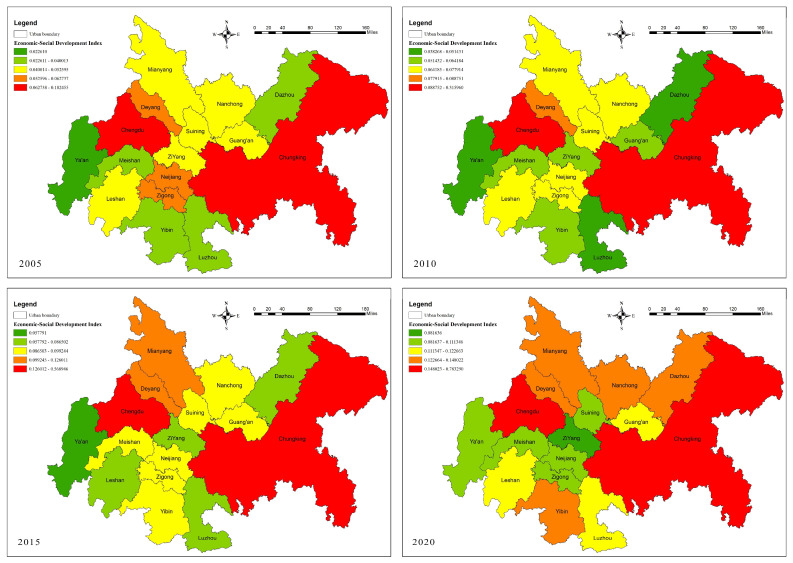
Changes in the economic–social development of cities in CCCG from 2005 to 2020.

**Figure 10 ijerph-20-05095-f010:**
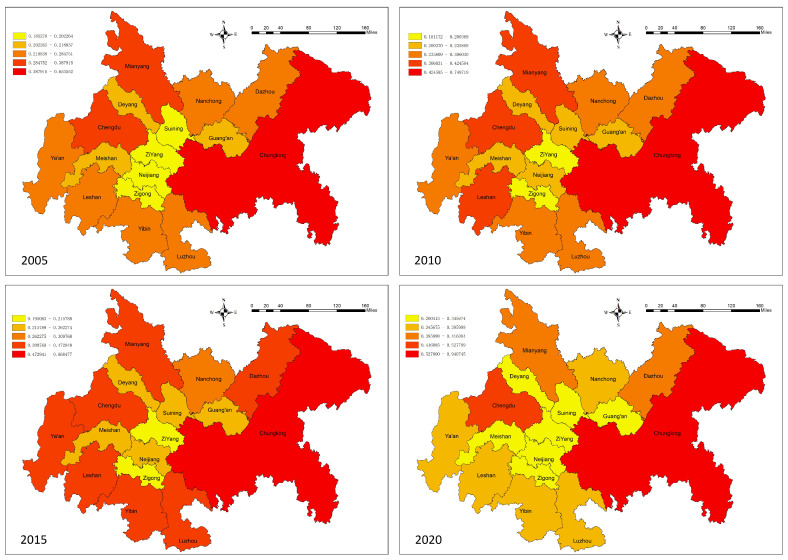
Spatial distribution of the level of coupling between economic–social development and ecosystem services in each city from 2005 to 2020.

**Figure 11 ijerph-20-05095-f011:**
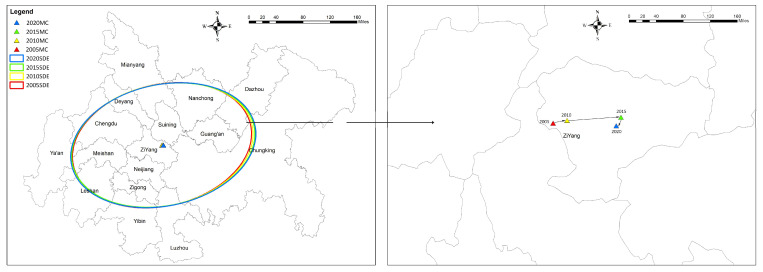
Coupling coordination standard deviation ellipse and center of gravity migration in CCCG, 2005–2020.

**Figure 12 ijerph-20-05095-f012:**
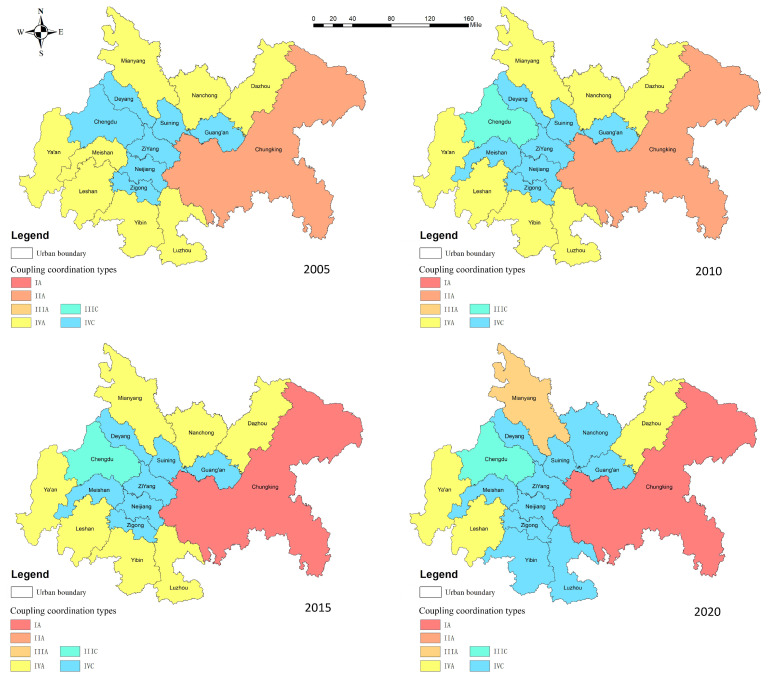
Distribution of the types of coupling and coordination between economic–social development and ecological service system in CCCG. Note: I is coordinated development type, II is transformation development type, III is running-in development type, IV is uncoordinated development type, A is economic lag type, and C is ecological lag type. For example, IA is a coordinated development of economic lag, and so on.

**Table 1 ijerph-20-05095-t001:** Indicators and weights of the economic–social system evaluation.

Indicators	Weights	Unit
Per capita GDP	0.0499	CNY
Proportion of tertiary industry in GDP	0.0403	%
Education Expenditure	0.2553	billion
Retail sales of social consumer hoods	0.1532	billion
Population density	0.0482	persons/km^2^
Final Number of Employees in Urban Units	0.2066	10,000 persons
Ratio of primary and secondary school students	0.0262	%
Number of industrial enterprises above the designated size	0.1174	/
Number of hospital beds per 10,000 people	0.1029	/

**Table 2 ijerph-20-05095-t002:** Value of ecosystem services per unit area of land in CCCG (Unit: CNY/hm^2^).

Primary Type	Secondary Type	Forest Land	Grass Land	Cultivated Land	Water Area	Construction Land	Desert
Supply service	Food production	681.68	888.25	2065.70	1838.47	0.00	41.31
	Raw material production	6155.78	743.65	805.62	1218.76	0.00	82.63
Conditioning service	Gas regulation	8923.82	3098.55	1487.30	6031.84	0.00	123.94
	Climate regulation	8407.39	3222.49	2003.73	32,245.55	0.00	268.54
	Hydrological regulation	8448.71	3139.86	1590.59	66,536.14	0.00	144.60
	Waste treatment	3553.00	2726.72	2871.32	60,421.68	0.00	537.08
Support services	Soil conservation	8304.11	4627.16	3036.58	4957.68	0.00	351.17
	Maintaining biodiversity	9316.30	3862.86	2107.01	14,707.77	0.00	826.28
Cultural services	Provide aesthetic landscape	4296.65	1797.16	351.17	18,859.83	0.00	495.77
	Total	58,087.44	24,106.70	16,319.02	20,6817.72	0.00	2871.32

**Table 3 ijerph-20-05095-t003:** Coupling degree of the coordination between land ecosystem services and economic–social development.

Serial	Coupling Coordination Index Range	Level
1	0–0.09	Extreme Disorder (Level 10: X)
2	0.1–0.19	Serious disorders (Level 9: IX)
3	0.2–0.29	Moderate disorder (Level 8: VIII)
4	0.3–0.39	Mild disorder (Level 7: VII)
5	0.4–0.49	Near Dysfunction (Level 6: VI)
6	0.5–0.59	Barely coordination (Level 5: V)
7	0.6–0.69	Primary coordination (Level 4: IV)
8	0.7–0.79	Intermediate coordination (Level 3: III)
9	0.8–0.89	Good coordination (Level 2: II)
10	0.9–1.0	High quality coordination (Level 1: I)

Note: The level of coordination degree can be divided into three subtypes according to the magnitude of E1 and E2: when E1 < E2, ecosystem service is lagging; when E1 = E2, land ecosystem services and social–economic development are in synchronization; when E1 > E2, social–economic development is lagging.

**Table 4 ijerph-20-05095-t004:** Value of services and their changes by ecosystem type, 2005–2020 (unit: billion).

	2005	2010	2015	2020	Variable Value	Average Variability	Average Contribution Rate
Cultivated land	2123.84	2113.79	2098.91	2081.02	−42.82	2.1%	26.55%
Forest land	4512.84	4629.72	4622.22	4667.53	154.69	3.3%	58.14%
Grassland	598.42	522.60	522.04	487.75	−110.67	22.7%	6.72%
Water area	615.19	670.10	691.84	744.18	128.99	17.3%	8.58%
Construction land	0.00	0.00	0.00	0.00	0.00	0.0%	0.00%
Desert	0.60	0.86	0.86	0.87	0.27	31.0%	0.01%
Total	7850.89	7937.07	7935.87	7981.35	130.46	76.4%	100.00%

**Table 5 ijerph-20-05095-t005:** Changes in the development of the elements of economic–social development in 2005–2020.

Indicator	2005	2010	2015	2020	Variable Value	Average Variability
Per capita GDP	139,259	333,694	597,064	866,335	727,076	44.49%
Proportion of tertiary industry in GDP	546.2	467.86	508.05	754.41	208.21	7.94%
Education Expenditure	2.66	39.54	115.57	1897.34	1894.68	84.32%
Retail sales of social consumer hoods	3746.46	8800.53	18,817.01	30,586.33	26,839.87	49.71%
Population density	8314.2	8582.9	8665.37	7499.36	−814.84	−3.82%
Final Number of Employees in Urban Units	641.55	726	1935.32	2027.46	1385.91	26.22%
Ratio of primary and secondary school students	0.77	0.91	1.04	1.11	0.34	11.39%
Number of industrial enterprises above the designated size	10,146	19,366	18,621	20,436	10,290	17.50%
Number of hospital beds per 10,000 people	218,519	33,6626	559,289	605,145	386,626	27.49%

**Table 6 ijerph-20-05095-t006:** Index of economic–social development level of cities in the Chengdu–Chongqing urban agglomeration, 2005–2020.

City	2005		2010		2015		2020	
	Score	Ranking	Score	Ranking	Score	Ranking	Score	Ranking
Chongqing	0.182	1	0.316	1	0.569	1	0.783	1
Chengdu	0.181	2	0.274	2	0.425	2	0.679	2
Neijiang	0.063	3	0.074	6	0.095	8	0.108	15
Deyang	0.062	4	0.089	3	0.126	3	0.135	7
Zigong	0.060	5	0.075	5	0.099	6	0.111	13
Nanchong	0.053	6	0.067	9	0.094	9	0.148	3
Mianyang	0.052	7	0.078	4	0.110	4	0.139	6
Guang’an	0.050	8	0.060	12	0.097	7	0.122	9
Suining	0.049	9	0.069	7	0.099	5	0.109	14
Leshan	0.044	10	0.067	8	0.087	12	0.118	10
ZiYang	0.044	11	0.064	10	0.086	13	0.082	16
Luzhou	0.040	12	0.051	14	0.078	15	0.123	8
Meishan	0.039	13	0.059	13	0.090	10	0.111	12
Yibin	0.038	14	0.061	11	0.090	11	0.142	5
Dazhou	0.036	15	0.050	15	0.080	14	0.143	4
Ya’an	0.023	16	0.038	16	0.058	16	0.111	11

**Table 7 ijerph-20-05095-t007:** Value of land ecosystem services and land-average value of land ecosystem services (units: billion CNY and CNY/hm^2^, respectively).

City	2005		2010		2015		2020	
	ESV	AESV	ESV	AESV	ESV	AESV	ESV	AESV
Chongqing	2885	34,964	2971	36,011	2969	35,983	3016	36,560
Chengdu	390	27,221	391	27,280	388	27,065	385	26,881
Neijiang	124	23,121	124	23,033	123	22,894	122	22,694
Deyang	158	26,781	154	26,028	152	25,816	153	25,947
Zigong	118	27,031	98	22,410	97	22,326	97	22,291
Nanchong	299	23,934	305	24,433	304	24,385	307	24,587
Mianyang	702	34,681	698	34,491	699	34,545	701	34,606
Guang’an	175	27,667	177	27,885	176	27,726	177	27,978
Suining	135	25,380	136	25,587	136	25,527	132	24,810
Leshan	479	37,610	478	37,518	478	37,504	482	37,821
ZiYang	121	20,974	120	20,851	119	20,727	120	20,871
Luzhou	418	34,190	421	34,411	420	34,347	416	34,047
Meishan	209	28,996	210	29,247	209	29,042	211	29,393
Yibin	428	32,304	431	32,526	432	32,588	434	32,728
Dazhou	570	34,323	574	34,560	574	34,564	567	34,130
Ya’an	612	40,806	621	41,423	631	42,066	632	42,095

**Table 8 ijerph-20-05095-t008:** Changes in the overall coupling coordination level of CCCG.

Year	2005	2010	2015	2020
Coupling coordination degree	0.50	0.55	0.61	0.66
Coupling coordination level	Barely coordinated (V)	Barely coordinated (V)	Primary coordination (IV)	Primary coordination (IV)

**Table 9 ijerph-20-05095-t009:** Levels of coupling between economic–social development and ecological service system in CCCG, 2005–2020.

City	2005		2010		2015		2020	
Chongqing	0.654	Primary Coordination (IV)	0.750	Intermediate Coordination (III)	0.868	Good Coordination (II)	0.941	High-quality Coordination (I)
Chengdu	0.388	Mild Disorder (VII)	0.425	Near Dysfunction (VI)	0.473	Near Dysfunction (VI)	0.528	Barely Coordinated (V)
Neijiang	0.202	Moderate Disorder (VIII)	0.207	Moderate Disorder (VIII)	0.221	Moderate Disorder (VIII)	0.227	Moderate Disorder (VIII)
Deyang	0.219	Moderate Disorder (VIII)	0.233	Moderate Disorder (VIII)	0.254	Moderate Disorder (VIII)	0.258	Moderate Disorder (VIII)
Zigong	0.188	Serious Disorder (IX)	0.181	Serious Disorder (IX)	0.195	Serious Disorder (IX)	0.200	Moderate Disorder (VIII)
Nanchong	0.270	Moderate Disorder (VIII)	0.285	Moderate Disorder (VIII)	0.310	Mild Disorder (VII)	0.346	Mild Disorder (VII)
Mianyang	0.328	Mild Disorder (VII)	0.360	Mild Disorder (VII)	0.392	Mild Disorder (VII)	0.416	Near Dysfunction (VI)
Guang’an	0.217	Moderate Disorder (VIII)	0.224	Moderate Disorder (VIII)	0.253	Moderate Disorder (VIII)	0.267	Moderate Disorder (VIII)
Suining	0.197	Serious Disorder (IX)	0.211	Moderate Disorder (VIII)	0.232	Moderate Disorder (VIII)	0.233	Moderate Disorder (VIII)
Leshan	0.284	Moderate Disorder (VIII)	0.311	Mild Disorder (VII)	0.331	Mild Disorder (VII)	0.357	Mild Disorder (VII)
ZiYang	0.185	Serious disorder (IX)	0.200	Moderate Disorder (VIII)	0.216	Moderate Disorder (VIII)	0.214	Moderate Disorder (VIII)
Luzhou	0.268	Moderate Disorder (VIII)	0.282	Moderate Disorder (VIII)	0.313	Mild Disorder (VII)	0.348	Mild Disorder (VII)
Meishan	0.214	Moderate Disorder (VIII)	0.236	Moderate Disorder (VIII)	0.262	Moderate Disorder (VIII)	0.276	Moderate Disorder (VIII)
Yibin	0.267	Moderate Disorder (VIII)	0.297	Moderate Disorder (VIII)	0.329	Mild Disorder (VII)	0.367	Mild Disorder (VII)
Dazhou	0.285	Moderate Disorder (VIII)	0.306	Mild Disorder (VII)	0.345	Mild Disorder (VII)	0.396	Mild Disorder (VII)
Ya’an	0.250	Moderate Disorder (VIII)	0.285	Moderate Disorder (VIII)	0.318	Mild Disorder (VII)	0.376	Mild Disorder (VII)

**Table 10 ijerph-20-05095-t010:** Ellipse of the standard deviation of coupling coordination and center of gravity migration parameters for the CCCG, 2005–2020.

Year	Ellipse Area (km^2^)	Ellipse Major Axis (km)	Ellipse Minor Axis (km)	Rotation Angle (°)	Center of Gravity Migration Distance (km)
2005	97,731.01	216.49	143.71	76.88	
2010	98,795.49	219.18	143.49	77.71	
2015	99,321.21	220.86	143.15	78.26	
2020	100,775.29	223.53	143.52	77.97	
2005–2010					0.67
2010–2015					2.53
2015–2020					0.31

## Data Availability

Not applicable.

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
