# Peer review of "Coupling Evaluation and Spatial–Temporal Evolution of Land Ecosystem Services and Economic–Social Development in a City Group: The Case Study of the Chengdu–Chongqing City Group"

_ijerph, 2023, doi:10.3390/ijerph20065095_

Round 1

Reviewer 1 Report

The paper presents a coupling coordination degree model of land use remote sensing data and socioeconomic development data in the case study of Chengdu-Chongquing City Group. It is a suitable topic for International Journal of Environmental Research and Public Health MDPI journal. I recommend a revision following my comments below.

GENERAL COMMENTS:

  • In the Abstract, the authors mention that this paper constructs a coupling evaluation model of economic-social development and land 13 ecosystem service. Nevertheless, this new model/method is not clearly described in the Methodology. What is the new method novelty? What is it build for?

  • Furthermore, the new model construction is not clearly mention in the Discussion and/or Conclusions.

  • The work’s novelty should be explored in the Abstract, Introduction, Discussion and Conclusion.

  • How could this new coupling evaluation model be applied elsewere?

SPECIFIC COMMENTS:

  • Abstract: The main goals of this paper and contributions to the literature should be highlighted. It is mostly focusing on results, which should also be mentioned, but better linked with the goals and future contributions.

  • Table 1: The sum of the weights is different of 1

  • Section 2.3.1: The weights obtention is not clear. They should be presented before normalization, with their sources and discussion.

  • Lines 189-190: This sentence is confusing. Please revise it.

  • The sections’ numbers should be corrected. There are two “2.3.1” sections.

  • Table 2: The values presented in Table 2 should be discussed.

  • Results: The results could be structured as an application of the Methodology to the case study. A brief introduction to each figure/table and a further explanation would be valuable.

  • Figures 1, 2, 6, 7, 8, 9, 10 and 11 are of low resolution.

Author Response

Dear reviewer:

Thanks very much for taking your time to review this manuscript. We appreciate all your generous comments and suggestions! Please find my itemized responses in below and my revisions in the re-submitted files.

Reviewer 2 Report

The paper is well organized, while, I would have to say, it is an interesting and good research topic in the cities of China. However, the author hasn’t provided a very rigorous research analysis framework and research mechanism analysis. Overall, the paper without any new findings and creatives. based on these, I have some concerns for the paper. My specific comments are in the bellowing:

(1) The abstract hasn’t been well organized, you should simply put forward you core research question, the theoretical bases, theoretical framework, and methodology, no need the literature review in the abstract. Some policy recommendation should include in the abstract.

(2)  The most serious problem is that the paper hasn’t formed a theoretical framework that can construct the relationship of land ecological service and the economic-social development, either the author can base on some theories to link these two factors, or the author can establish the conceptual framework to construct the influence relationship between each other.

(3) 2.3.1 has a very serious problem of the title, it should be Assessment of the level of land ecological system service. The authors should not be so careless.

(4)  The most part of the paper are doing a job: assessment and calculation, just use some method to calculate the value of the economic-social development and land ecological system value, while without any mechanism analysis, which are the factors leads to the differences and tendency of these two systems.

(5)  In the sections 3.3, the authors are doing the coordination calculation of the different cities in the research areas, and has ranked all of the cities, and given a very obvious of the results to readers, we are suspecting the analysis method/calculation methods first, then we are wondering why the authors do not do some mechanism analysis, what are the reasons resulted in such difference in the coordination distribution, if these aspects are more interesting.  

(6) The discussion sections should be simpler, some of the results have been presented in the results sections, so the contents in the part should be simpler and just present the main contents with some sentences.

(7)  The research limitations should put after the conclusion, and should not put in the discussion.

(8) Anther serious problems lies in that the authors have summarized the research findings and listed them in the conclusion, but without any policy implications. For such paper to research the relationships of the land system service and economic-social development in some specific areas, the policy suggestions and implications for the local authorizes are very important and very significant, the author should put some paragraphs to provide specific policy suggestion.   

Author Response

(The authors gave the same response as above.)

Round 2

Reviewer 1 Report

It is noted that the revised version of the manuscript displays significant improvements and has successfully addressed the concerns raised in the initial review.

The authors have taken into consideration the feedback provided in the first review and have made substantial revisions to the manuscript. The revised version now presents a clearer and more coherent argument, and the overall organization of the paper has been improved.

In addition, more detailed and comprehensive explanations of the methodology have been provided by the authors, which has helped to clarify the research design and the data analysis process. The results have also been presented more clearly, with relevant graphs and tables to support the findings.

Author Response

Dear reviewer:

Thanks very much for taking your time to review this manuscript. We appreciate all your generous comments and suggestions! Please find my itemized responses in below and my revisions in the re-submitted files.

Sincerely yours,

Qikang Zhong, Zhe Li, Yujing He

Reviewer 2 Report

The paper is well organized, I can find the authors have done great efforts for the modification of the paper, and the paper has been much improved. Overall, the paper has been better than before. My comments are in the bellowing:

(1)  I agree with the authors, land ecosystem services formed by the ecosystem to maintain human survival and development, mainly including supply services, regulation services, support services, and cultural services, while, the author should also consider if there are some mechanisms that the economic-social development may have impacts on land ecological service through some different mechanisms?

(2)  In my opinion, the authors have presented the research area in Chengdu and Chongqing areas, and the land ecological services constitution, while the author should present how does the research areas located in the whole picture of China?  

(3) The figure 12 in the paper is much messy and it has no legend in the map, also some information in the map is not sufficient.

(4)  The policy recommendation in the conclusion is not sufficient. Actually, the author can give and provide the suggestions according to their results in the section 3, and then from different dimensions, and list them as (1), (2), (3) …., these will be more acceptable and more complete.  

Author Response

(The authors gave the same response as above.)
